# On Elicitation Complexity

**Rafael Frongillo**
University of Colorado, Boulder
raf@colorado.edu

**Ian A. Kash**
Microsoft Research
iankash@microsoft.com

## Abstract

Elicitation is the study of statistics or *properties* which are computable via empirical risk minimization. While several recent papers have approached the general question of which properties are elicitable, we suggest that this is the wrong question—all properties are elicitable by first eliciting the entire distribution or data set, and thus the important question is *how* elicitable. Specifically, what is the minimum number of regression parameters needed to compute the property?

Building on previous work, we introduce a new notion of elicitation complexity and lay the foundations for a calculus of elicitation. We establish several general results and techniques for proving upper and lower bounds on elicitation complexity. These results provide tight bounds for eliciting the Bayes risk of any loss, a large class of properties which includes spectral risk measures and several new properties of interest.

## 1 Introduction

Empirical risk minimization (ERM) is a dominant framework for supervised machine learning, and a key component of many learning algorithms. A statistic or *property* is simply a functional assigning a vector of values to each distribution. We say that such a property is *elicitable*, if for some loss function it can be represented as the unique minimizer of the expected loss under the distribution. Thus, the study of which properties are elicitable can be viewed as the study of which statistics are computable via ERM [1, 2, 3].

The study of property elicitation began in statistics [4, 5, 6, 7], and is gaining momentum in machine learning [8, 1, 2, 3], economics [9, 10], and most recently, finance [11, 12, 13, 14, 15]. A sequence of papers starting with Savage [4] has looked at the full characterization of losses which elicit the mean of a distribution, or more generally the expectation of a vector-valued random variable [16, 3]. The case of real-valued properties is also now well in hand [9, 1]. The general vector-valued case is still generally open, with recent progress in [3, 2, 15]. Recently, a parallel thread of research has been underway in finance, to understand which financial risk measures, among several in use or proposed to help regulate the risks of financial institutions, are computable via regression, i.e., elicitable (cf. references above). More often than not, these papers have concluded that most risk measures under consideration are not elicitable, notable exceptions being generalized quantiles (e.g. value-at-risk, expectiles) and expected utility [13, 12].

Throughout the growing momentum of the study of elicitation, one question has been central: which properties are elicitable? It is clear, however, that all properties are "indirectly" elicitable if one first elicits the distribution using a standard proper scoring rule. Therefore, in the present work, we suggest replacing this question with a more nuanced one: *how* elicitable are various properties? Specifically, heeding the suggestion of Gneiting [7], we adapt to our setting the notion of *elicitation complexity* introduced by Lambert et al. [17], which captures how many parameters one needs to maintain in an ERM procedure for the property in question. Indeed, if a real-valued property is found not to be elicitable, such as the variance, one should not abandon it, but rather ask how many parameters are required to compute it via ERM.

Our work is heavily inspired by the recent progress along these lines of Fissler and Ziegel [15], who show that spectral risk measures of support $k$ have elicitation complexity at most $k + 1$. Spectral risk measures are among those under consideration in the finance community, and this result shows that while not elicitable in the classical sense, their elicitation complexity is still low, and hence one can develop reasonable regression procedures for them. Our results extend to these and many other risk measures (see § 4.6), often providing matching *lower bounds* on the complexity as well.

Our contributions are the following. We first introduce an adapted definition of elicitation complexity which we believe to be the right notion to focus on going forward. We establish a few simple but useful results which allow for a kind of calculus of elicitation; for example, conditions under which the complexity of eliciting two properties in tandem is the sum of their individual complexities. In § 3, we derive several techniques for proving both upper and lower bounds on elicitation complexity which apply primarily to the *Bayes risks* from decision theory, or optimal expected loss functions. The class includes spectral risk measures among several others; see § 4. We conclude with brief remarks and open questions.

## 2 Preliminaries and Foundation

Let $\Omega$ be a set of outcomes and $\mathcal{P} \subseteq \Delta(\Omega)$ be a convex set of probability measures. The goal of elicitation is to learn something about the distribution $p \in \mathcal{P}$, specifically some function $\Gamma(p)$ such as the mean or variance, by minimizing a loss function.

**Definition 1.** *A* property *is a function* $\Gamma : \mathcal{P} \to \mathbb{R}^k$, *for some* $k \in \mathbb{N}$, *which associates a desired report value to each distribution.*[1] *We let* $\Gamma_r \doteq \{p \in \mathcal{P} \,|\, r = \Gamma(p)\}$ *denote the set of distributions* $p$ *corresponding to report value* $r$.

Given a property $\Gamma$, we want to ensure that the best result is to reveal the value of the property using a *loss function* that evaluates the report using a sample from the distribution.

**Definition 2.** *A* loss function $L : \mathbb{R}^k \times \Omega \to \mathbb{R}$ elicits *a property* $\Gamma : \mathcal{P} \to \mathbb{R}^k$ *if for all* $p \in \mathcal{P}$, $\Gamma(p) = \operatorname{arginf}_r L(r, p)$, *where* $L(r, p) \doteq \mathbb{E}_p[L(r, \cdot)]$. *A property is* elicitable *if some loss elicits it.*

For example, when $\Omega = \mathbb{R}$, the mean $\Gamma(p) = \mathbb{E}_p[\omega]$ is elicitable via squared loss $L(r, \omega) = (r - \omega)^2$.

A well-known necessary condition for elicitability is convexity of the level sets of $\Gamma$.

**Proposition 1** (Osband [5]). *If* $\Gamma$ *is elicitable, the level sets* $\Gamma_r$ *are convex for all* $r \in \Gamma(\mathcal{P})$.

One can easily check that the mean $\Gamma(p) = \mathbb{E}_p[\omega]$ has convex level sets, yet the variance $\Gamma(p) = \mathbb{E}_p[(\omega - \mathbb{E}_p[\omega])^2]$ does not, and hence is not elicitable [9].

It is often useful to work with a stronger condition, that not only is $\Gamma_r$ convex, but it is the intersection of a linear subspace with $\mathcal{P}$. This condition is equivalent the existence of an *identification function*, a functional describing the level sets of $\Gamma$ [17, 1].

**Definition 3.** *A function* $V : \mathcal{R} \times \Omega \to \mathbb{R}^k$ *is an* identification function *for* $\Gamma : \mathcal{P} \to \mathbb{R}^k$, *or* identifies $\Gamma$, *if for all* $r \in \Gamma(\mathcal{P})$ *it holds that* $p \in \Gamma_r \iff V(r, p) = 0 \in \mathbb{R}^k$, *where as with* $L(r, p)$ *above we write* $V(r, p) \doteq \mathbb{E}_p[V(r, \omega)]$. $\Gamma$ *is* identifiable *if there exists a* $V$ *identifying it.*

One can check for example that $V(r, \omega) = \omega - r$ identifies the mean.

We can now define the classes of identifiable and elicitable properties, along with the complexity of identifying or eliciting a given property. Naturally, a property is $k$-identifiable if it is the link of a $k$-dimensional identifiable property, and $k$-elicitable if it is the link of a $k$-dimensional elicitable property. The elicitation complexity of a property is then simply the minimum dimension $k$ needed for it to be $k$-elicitable.

**Definition 4.** *Let* $\mathcal{I}_k(\mathcal{P})$ *denote the class of all identifiable properties* $\Gamma : \mathcal{P} \to \mathbb{R}^k$, *and* $\mathcal{E}_k(\mathcal{P})$ *denote the class of all elicitable properties* $\Gamma : \mathcal{P} \to \mathbb{R}^k$. *We write* $\mathcal{I}(\mathcal{P}) = \bigcup_{k \in \mathbb{N}} \mathcal{I}_k(\mathcal{P})$ *and* $\mathcal{E}(\mathcal{P}) = \bigcup_{k \in \mathbb{N}} \mathcal{E}_k(\mathcal{P})$.

**Definition 5.** *A property* $\Gamma$ *is* $k$-identifiable *if there exists* $\hat{\Gamma} \in \mathcal{I}_k(\mathcal{P})$ *and* $f$ *such that* $\Gamma = f \circ \hat{\Gamma}$. *The* identification complexity *of* $\Gamma$ *is defined as* $\operatorname{iden}(\Gamma) = \min\{k : \Gamma \text{ is } k\text{-identifiable}\}$.

**Definition 6.** *A property $\Gamma$ is $k$-elicitable if there exists $\hat{\Gamma} \in \mathcal{E}_k(\mathcal{P})$ and $f$ such that $\Gamma = f \circ \hat{\Gamma}$. The elicitation complexity of $\Gamma$ is defined as* $\mathsf{elic}(\Gamma) = \min\{k : \Gamma \text{ is } k\text{-elicitable}\}$.

To make the above definitions concrete, recall that the variance $\sigma^2(p) = \mathbb{E}_p[(\mathbb{E}_p[\omega] - \omega)^2]$ is not elicitable, as its level sets are not convex, a necessary condition by Prop. 1. Note however that we may write $\sigma^2(p) = \mathbb{E}_p[\omega^2] - \mathbb{E}_p[\omega]^2$, which can be obtained from the property $\hat{\Gamma}(p) = (\mathbb{E}_p[\omega], \mathbb{E}_p[\omega^2])$. It is well-known [4, 7] that $\hat{\Gamma}$ is both elicitable and identifiable as the expectation of a vector-valued random variable $X(\omega) = (\omega, \omega^2)$, using for example $L(r, \omega) = \|r - X(\omega)\|^2$ and $V(r, \omega) = r - X(\omega)$. Thus, we can recover $\sigma^2$ as a link of the elicitable and identifiable $\hat{\Gamma} : \mathcal{P} \to \mathbb{R}^2$, and as no such $\hat{\Gamma} : \mathcal{P} \to \mathbb{R}$ exists, we have $\mathsf{iden}(\sigma^2) = \mathsf{elic}(\sigma^2) = 2$.

In this example, the variance has a stronger property than merely being 2-identifiable and 2-elicitable, namely that there is a single $\hat{\Gamma}$ that satisfies both of these simultaneously. In fact this is quite common, and identifiability provides geometric structure that we make use of in our lower bounds. Thus, most of our results use this refined notion of elicitation complexity.

**Definition 7.** *A property $\Gamma$ has* (identifiable) elicitation complexity
$\mathsf{elic}_{\mathcal{I}}(\Gamma) = \min\{k : \exists \hat{\Gamma}, f \text{ such that } \hat{\Gamma} \in \mathcal{E}_k(\mathcal{P}) \cap \mathcal{I}_k(\mathcal{P}) \text{ and } \Gamma = f \circ \hat{\Gamma}\}$.

Note that restricting our attention to $\mathsf{elic}_{\mathcal{I}}$ effectively requires $\mathsf{elic}_{\mathcal{I}}(\Gamma) \geq \mathsf{iden}(\Gamma)$; specifically, if $\Gamma$ is derived from some elicitable $\hat{\Gamma}$, then $\hat{\Gamma}$ must be identifiable as well. This restriction is only relevant for our lower bounds, as our upper bounds give losses explicitly.[2] Note however that *some* restriction on $\mathcal{E}_k(\mathcal{P})$ is necessary, as otherwise pathological constructions giving injective mappings from $\mathbb{R}$ to $\mathbb{R}^k$ would render all properties 1-elicitable. To alleviate this issue, some authors require continuity (e.g. [1]) while others like we do require identifiability (e.g. [15]), which can be motivated by the fact that for any differentiable loss $L$ for $\Gamma$, $V(r, \omega) = \nabla_r L(\cdot, \omega)$ will identify $\Gamma$ provided $\mathbb{E}_p[L]$ has no inflection points or local minima. An important future direction is to relax this identifiability assumption, as there are very natural (set-valued) properties with $\mathsf{iden} > \mathsf{elic}$.[3]

Our definition of elicitation complexity differs from the notion proposed by Lambert et al. [17], in that the components of $\hat{\Gamma}$ above do not need to be individually elicitable. This turns out to have a large impact, as under their definition the property $\Gamma(p) = \max_{\omega \in \Omega} p(\{\omega\})$ for finite $\Omega$ has elicitation complexity $|\Omega| - 1$, whereas under our definition $\mathsf{elic}_{\mathcal{I}}(\Gamma) = 2$; see Example 4.3. Fissler and Ziegel [15] propose a closer but still different definition, with the complexity being the smallest $k$ such that $\Gamma$ is a component of a $k$-dimensional elicitable property. Again, this definition can lead to larger complexities than necessary; take for example the squared mean $\Gamma(p) = \mathbb{E}_p[\omega]^2$ when $\Omega = \mathbb{R}$, which has $\mathsf{elic}_{\mathcal{I}}(\Gamma) = 1$ with $\hat{\Gamma}(p) = \mathbb{E}_p[\omega]$ and $f(x) = x^2$, but is not elicitable and thus has complexity 2 under [15]. We believe that, modulo regularity assumptions on $\mathcal{E}_k(\mathcal{P})$, our definition is better suited to studying the difficulty of eliciting properties: viewing $f$ as a (potentially dimension-reducing) link function, our definition captures the minimum number of parameters needed in an ERM computation of the property in question, followed by a simple one-time application of $f$.

## 2.1 Foundations of Elicitation Complexity

In the remainder of this section, we make some simple, but useful, observations about $\mathsf{iden}(\Gamma)$ and $\mathsf{elic}_{\mathcal{I}}(\Gamma)$. We have already discussed one such observation after Definition 7: $\mathsf{elic}_{\mathcal{I}}(\Gamma) \geq \mathsf{iden}(\Gamma)$.

It is natural to start with some trivial upper bounds. Clearly, whenever $p \in \mathcal{P}$ can be uniquely determined by some number of elicitable parameters then the elicitation complexity of every property is at most that number. The following propositions give two notable applications of this observation.[4]

**Proposition 2.** *When $|\Omega| = n$, every property $\Gamma$ has* $\mathsf{elic}_{\mathcal{I}}(\Gamma) \leq n - 1$.

*Proof.* The probability distribution is determined by the probability of any $n - 1$ outcomes, and the probability associated with a given outcome is both elicitable and identifiable. $\qquad\square$

**Proposition 3.** *When* $\Omega = \mathbb{R}$,[5] *every property* $\Gamma$ *has* $\mathsf{elic}_{\mathcal{I}}(\Gamma) \leq \boldsymbol{\omega}$ *(countable).*[6]

One well-studied class of properties are those where $\Gamma$ is linear, i.e., the expectation of some vector-valued random variable. All such properties are elicitable and identifiable (cf. [4, 8, 3]), with $\mathsf{elic}_{\mathcal{I}}(\Gamma) \leq k$, but of course the complexity can be lower if the range of $\Gamma$ is not full-dimensional.

**Lemma 1.** *Let* $X : \Omega \to \mathbb{R}^k$ *be* $\mathcal{P}$*-integrable and* $\Gamma(p) = \mathbb{E}_p[X]$. *Then* $\mathsf{elic}_{\mathcal{I}}(\Gamma) = \dim(\mathrm{affhull}(\Gamma(\mathcal{P})))$, *the dimension of the affine hull of the range of* $\Gamma$.

It is easy to create redundant properties in various ways. For example, given elicitable properties $\Gamma_1$ and $\Gamma_2$ the property $\Gamma \doteq \{\Gamma_1, \Gamma_2, \Gamma_1 + \Gamma_2\}$ clearly contains redundant information. A concrete case is $\Gamma = \{\text{mean squared, variance, 2nd moment}\}$, which, as we have seen, has $\mathsf{elic}_{\mathcal{I}}(\Gamma) = 2$. The following definitions and lemma capture various aspects of a lack of such redundancy.

**Definition 8.** *Property* $\Gamma : \mathcal{P} \to \mathbb{R}^k$ *in* $\mathcal{I}(\mathcal{P})$ *is of* full rank *if* $\mathsf{iden}(\Gamma) = k$.

Note that there are two ways for a property to fail to be full rank. First, as the examples above suggest, $\Gamma$ can be "redundant" so that it is a link of a lower-dimensional identifiable property. Full rank can also be violated if *more* dimensions are needed to identify the property than to specify it. This is the case with, e.g., the variance which is a 1 dimensional property but has $\mathsf{iden}(\sigma^2) = 2$.

**Definition 9.** *Properties* $\Gamma, \Gamma' \in \mathcal{I}(\mathcal{P})$ *are* independent *if* $\mathsf{iden}(\{\Gamma, \Gamma'\}) = \mathsf{iden}(\Gamma) + \mathsf{iden}(\Gamma')$.

**Lemma 2.** *If* $\Gamma, \Gamma' \in \mathcal{E}(\mathcal{P})$ *are full rank and independent, then* $\mathsf{elic}_{\mathcal{I}}(\{\Gamma, \Gamma'\}) = \mathsf{elic}_{\mathcal{I}}(\Gamma) + \mathsf{elic}_{\mathcal{I}}(\Gamma')$.

To illustrate the lemma, $\mathsf{elic}_{\mathcal{I}}(\text{variance}) = 2$, yet $\Gamma = \{\text{mean,variance}\}$ has $\mathsf{elic}_{\mathcal{I}}(\Gamma) = 2$, so clearly the mean and variance are not both independent and full rank. (As we have seen, variance is not full rank.) However, the mean and second moment satisfy both by Lemma 1.

Another important case is when $\Gamma$ consists of some number of distinct quantiles. Osband [5] essentially showed that quantiles are independent and of full rank, so their elicitation complexity is the number of quantiles being elicited.

**Lemma 3.** *Let* $\Omega = \mathbb{R}$ *and* $\mathcal{P}$ *be a class of probability measures with continuously differentiable and invertible CDFs* $F$, *which is sufficiently rich in the sense that for all* $x_1, \ldots, x_k \in \mathbb{R}$, $\mathrm{span}(\{F^{-1}(x_1), \ldots, F^{-1}(x_k)\}, F \in \mathcal{P}) = \mathbb{R}^k$. *Let* $q_\alpha$, *denote the* $\alpha$*-quantile function. Then if* $\alpha_1, \ldots, \alpha_k$ *are all distinct,* $\Gamma = \{q_{\alpha_1}, \ldots, q_{\alpha_k}\}$ *has* $\mathsf{elic}_{\mathcal{I}}(\Gamma) = k$.

The quantile example in particular allows us to see that all complexity classes, including $\boldsymbol{\omega}$, are occupied. In fact, our results to follow will show something stronger: even for *real-valued* properties $\Gamma : \mathcal{P} \to \mathbb{R}$, all classes are occupied; we give here the result that follows from our bounds on spectral risk measures in Example 4.4, but this holds for many other $\mathcal{P}$; see e.g. Example 4.2.

**Proposition 4.** *Let* $\mathcal{P}$ *as in Lemma 3. Then for all* $k \in \mathbb{N}$ *there exists* $\gamma : \mathcal{P} \to \mathbb{R}$ *with* $\mathsf{elic}_{\mathcal{I}}(\gamma) = k$.

## 3 Eliciting the Bayes Risk

In this section we prove two theorems that provide our main tools for proving upper and lower bounds respectively on elicitation complexity. Of course many properties are known to be elicitable, and the losses that elicit them provide such an upper bound for that case. We provide such a construction for properties that can be expressed as the pointwise minimum of an indexed set of functions. Interestingly, our construction does not elicit the minimum directly, but as a joint elicitation of the value and the function that realizes this value. The form (1) is that of a scoring rule for the linear property $p \mapsto \mathbb{E}_p[X_a]$, except that here the index $a$ itself is also elicited.[7]

**Theorem 1.** *Let* $\{X_a : \Omega \to \mathbb{R}\}_{a \in \mathcal{A}}$ *be a set of* $\mathcal{P}$*-integrable functions indexed by* $\mathcal{A} \subseteq \mathbb{R}^k$. *Then if* $\inf_a \mathbb{E}_p[X_a]$ *is attained, the property* $\gamma(p) = \min_a \mathbb{E}_p[X_a]$ *is* $(k+1)$*-elicitable. In particular,*

$$L((r, a), \omega) = H(r) + h(r)(X_a - r) \tag{1}$$

*elicits* $p \mapsto \{(\gamma(p), a) : \mathbb{E}_p[X_a] = \gamma(p)\}$ *for any strictly decreasing* $h : \mathbb{R} \to \mathbb{R}_+$ *with* $\frac{d}{dr}H = h$.

*Proof.* We will work with gains instead of losses, and show that $S((r, a), \omega) = g(r) + dg_r(X_a - r)$ elicits $p \mapsto \{(\gamma(p), a) : \mathbb{E}_p[X_a] = \gamma(p)\}$ for $\gamma(p) = \max_a \mathbb{E}_p[X_a]$. Here $g$ is convex with strictly increasing and positive subgradient $dg$.

For any fixed $a$, we have by the subgradient inequality,

$$S((r, a), p) = g(r) + dg_r(\mathbb{E}_p[X_a] - r) \leq g(\mathbb{E}_p[X_a]) = S((\mathbb{E}_p[X_a], a), p) \ ,$$

and as $dg$ is strictly increasing, $g$ is strictly convex, so $r = \mathbb{E}_p[X_a]$ is the unique maximizer. Now letting $\tilde{S}(a, p) = S((\mathbb{E}_p[X_a], a), p)$, we have

$$\operatorname*{argmax}_{a \in \mathcal{A}} \tilde{S}(a, p) = \operatorname*{argmax}_{a \in \mathcal{A}} g(\mathbb{E}_p[X_a]) = \operatorname*{argmax}_{a \in \mathcal{A}} \mathbb{E}_p[X_a] \ ,$$

because $g$ is strictly increasing. We now have

$$\operatorname*{argmax}_{a \in \mathcal{A}, r \in \mathbb{R}} S((r, a), p) = \left\{ (\mathbb{E}_p[X_a], a) : a \in \operatorname*{argmax}_{a \in \mathcal{A}} \mathbb{E}_p[X_a] \right\} \ . \qquad \square$$

One natural way to get such an indexed set of functions is to take an arbitrary loss function $L(r, \omega)$, in which case this pointwise minimum corresponds to the *Bayes risk*, which is simply the minimum possible expected loss under some distribution $p$.

**Definition 10.** *Given loss function $L : \mathcal{A} \times \Omega \to \mathbb{R}$ on some prediction set $\mathcal{A}$, the* Bayes risk *of $L$ is defined as $\underline{L}(p) := \inf_{a \in \mathcal{A}} L(a, p)$.*

One illustration of the power of Theorem 1 is that the Bayes risk of a loss eliciting a $k$-dimensional property is itself $(k+1)$-elicitable.

**Corollary 1.** *If $L : \mathbb{R}^k \times \Omega \to \mathbb{R}$ is a loss function eliciting $\Gamma : \mathcal{P} \to \mathbb{R}^k$, then the loss*

$$L((r, a), \omega) = L'(a, \omega) + H(r) + h(r)(L(a, \omega) - r) \tag{2}$$

*elicits $\{\underline{L}, \Gamma\}$, where $h : \mathbb{R} \to \mathbb{R}_+$ is any positive strictly decreasing function, $H(r) = \int_0^r h(x) dx$, and $L'$ is any surrogate loss eliciting $\Gamma$.[8] If $\Gamma \in \mathcal{I}_k(\mathcal{P})$, $\mathsf{elic}_\mathcal{I}(\underline{L}) \leq k + 1$.*

We now turn to our second theorem which provides lower bounds for the elicitation complexity of the Bayes risk. A first observation, which follows from standard convex analysis, is that $\underline{L}$ is concave, and thus it is unlikely to be elicitable directly, as the level sets of $\underline{L}$ are likely to be non-convex. To show a lower bound greater than 1, however, we will need much stronger techniques. In particular, while $\underline{L}$ must be concave, it may not be strictly so, thus enabling level sets which are potentially amenable to elicitation. In fact, $\underline{L}$ *must* be flat between any two distributions which share a minimizer. Crucial to our lower bound is the fact that whenever the minimizer of $L$ *differs* between two distributions, $\underline{L}$ is essentially strictly concave between them.

**Lemma 4.** *Suppose loss $L$ with Bayes risk $\underline{L}$ elicits $\Gamma : \mathcal{P} \to \mathbb{R}^k$. Then for any $p, p' \in \mathcal{P}$ with $\Gamma(p) \neq \Gamma(p')$, we have $\underline{L}(\lambda p + (1 - \lambda)p') > \lambda \underline{L}(p) + (1 - \lambda)\underline{L}(p')$ for all $\lambda \in (0, 1)$.*

With this lemma in hand we can prove our lower bound. The crucial insight is that an identification function for the Bayes risk of a loss eliciting a property can, through a link, be used to identify that property. Corollary 1 tells us that $k + 1$ parameters suffice for the Bayes risk of a $k$-dimensional property, and our lower bound shows this is often necessary. Only $k$ parameters suffice, however, when the property value itself provides all the information required to compute the Bayes risk; for example, dropping the $y^2$ term from squared loss gives $L(x, y) = x^2 - 2xy$ and $\underline{L}(p) = -\mathbb{E}_p[y]^2$, giving $\mathsf{elic}(\underline{L}) = 1$. Thus the theorem splits the lower bound into two cases.

**Theorem 2.** *If a loss $L$ elicits some $\Gamma \in \mathcal{E}_k(\mathcal{P})$ with elicitation complexity $\mathsf{elic}_\mathcal{I}(\Gamma) = k$, then its Bayes risk $\underline{L}$ has $\mathsf{elic}_\mathcal{I}(\underline{L}) \geq k$. Moreover, if we can write $\underline{L} = f \circ \Gamma$ for some function $f : \mathbb{R}^k \to \mathbb{R}$, then we have $\mathsf{elic}_\mathcal{I}(\underline{L}) = k$; otherwise, $\mathsf{elic}_\mathcal{I}(\underline{L}) = k + 1$.*

*Proof.* Let $\hat{\Gamma} \in \mathcal{E}_\ell$ such that $\underline{L} = g \circ \hat{\Gamma}$ for some $g : \mathbb{R}^\ell \to \mathbb{R}$.

We show by contradiction that for all $p, p' \in \mathcal{P}$, $\hat{\Gamma}(p) = \hat{\Gamma}(p')$ implies $\Gamma(p) = \Gamma(p')$. Otherwise, we have $p, p'$ with $\hat{\Gamma}(p) = \hat{\Gamma}(p')$, and thus $\underline{L}(p) = \underline{L}(p')$, but $\Gamma(p) \neq \Gamma(p')$. Lemma 4 would then give us some $p_\lambda = \lambda p + (1 - \lambda)p'$ with $\underline{L}(p_\lambda) > \underline{L}(p)$. But as the level sets $\hat{\Gamma}_{\hat{r}}$ are convex by Prop. 1, we would have $\hat{\Gamma}(p_\lambda) = \hat{\Gamma}(p)$, which would imply $\underline{L}(p_\lambda) = \underline{L}(p)$.

We now can conclude that there exists $h : \mathbb{R}^\ell \to \mathbb{R}^k$ such that $\Gamma = h \circ \hat{\Gamma}$. But as $\hat{\Gamma} \in \mathcal{E}_\ell$, this implies $\mathsf{elic}_\mathcal{I}(\Gamma) \leq \ell$, so clearly we need $\ell \geq k$. Finally, if $\ell = k$ we have $\underline{L} = g \circ \hat{\Gamma} = g \circ h^{-1} \circ \Gamma$. The upper bounds follow from Corollary 1. $\qquad\square$

# 4    Examples and Applications

We now give several applications of our results. Several upper bounds are novel, as well as all lower bounds greater than 2. In the examples, unless we refer to $\Omega$ explicitly we will assume $\Omega = \mathbb{R}$ and write $y \in \Omega$ so that $y \sim p$. In each setting, we also make several standard regularity assumptions which we suppress for ease of exposition — for example, for the variance and variantile we assume finite first and second moments (which must span $\mathbb{R}^2$), and whenever we discuss quantiles we will assume that $\mathcal{P}$ is as in Lemma 3, though we will not require as much regularity for our upper bounds.

## 4.1    Variance

In Section 2 we showed that $\mathsf{elic}_\mathcal{I}(\sigma^2) = 2$. As a warm up, let us see how to recover this statement using our results on the Bayes risk. We can view $\sigma^2$ as the Bayes risk of squared loss $L(x, y) = (x - y)^2$, which of course elicits the mean: $\underline{L}(p) = \min_{x \in \mathbb{R}} \mathbb{E}_p[(x - y)^2] = \mathbb{E}_p[(\mathbb{E}_p[y] - y)^2] = \sigma^2(p)$. This gives us $\mathsf{elic}_\mathcal{I}(\sigma^2) \leq 2$ by Corollary 1, with a matching lower bound by Theorem 2, as the variance is not simply a function of the mean. Corollary 1 gives losses such as $L((x, v), y) = e^{-v}((x - y)^2 - v) - e^{-v}$ which elict $\{\mathbb{E}_p[y], \sigma^2(p)\}$, but in fact there are losses which cannot be represented by the form (2), showing that we do not have a full characterization; for example, $\hat{L}((x, v), y) = v^2 + v(x - y)(2(x + y) + 1) + (x - y)^2 \left((x + y)^2 + x + y + 1\right)$. This $\hat{L}$ was generated via squared loss $\left\| z - \begin{bmatrix} y \\ y^2 \end{bmatrix} \right\|^2$ with respect to the norm $\|z\|^2 = z^\top \begin{bmatrix} 1 & -1/2 \\ -1/2 & 1 \end{bmatrix} z$, which elicits the first two moments, and link function $(z_1, z_2) \mapsto (z_1, z_2 - z_1^2)$.

## 4.2    Convex Functions of Means

Another simple example is $\gamma(p) = G(\mathbb{E}_p[X])$ for some strictly convex function $G : \mathbb{R}^k \to \mathbb{R}$ and $\mathcal{P}$-integrable $X : \Omega \to \mathbb{R}^k$. To avoid degeneracies, we assume $\dim \mathrm{affhull}\{\mathbb{E}_p[X] : p \in \mathcal{P}\} = k$, i.e. $\Gamma$ is full rank. Letting $\{dG_p\}_{p \in \mathcal{P}}$ be a selection of subgradients of $G$, the loss $L(r, \omega) = -(G(r) + dG_r(X(\omega) - r))$ elicits $\Gamma : p \mapsto \mathbb{E}_p[X]$ (cf. [3]), and moreover we have $\gamma(p) = -\underline{L}(p)$. By Lemma 1, $\mathsf{elic}_\mathcal{I}(\Gamma) = k$. One easily checks that $\underline{L} = G \circ \Gamma$, so now by Theorem 2, $\mathsf{elic}_\mathcal{I}(\gamma) = k$ as well. Letting $\{X_k\}_{k \in \mathbb{N}}$ be a family of such "full rank" random variables, this gives us a sequence of real-valued properties $\gamma_k(p) = \|\mathbb{E}_p[X]\|^2$ with $\mathsf{elic}_\mathcal{I}(\gamma_k) = k$, proving Proposition 4.

## 4.3    Modal Mass

With $\Omega = \mathbb{R}$ consider the property $\gamma_\beta(p) = \max_{x \in \mathbb{R}} p([x - \beta, x + \beta])$, namely, the maximum probability mass contained in an interval of width $2\beta$. Theorem 1 easily shows $\mathsf{elic}_\mathcal{I}(\gamma_\beta) \leq 2$, as $\hat{\gamma}_\beta(p) = \mathrm{argmax}_{x \in \mathbb{R}} p([x - \beta, x + \beta])$ is elicited by $L(x, y) = \mathbb{1}_{|x - y| > \beta}$, and $\gamma_\beta(p) = 1 - \underline{L}(p)$. Similarly, in the case of finite $\Omega$, $\gamma(p) = \max_{\omega \in \Omega} p(\{\omega\})$ is simply the expected score (gain rather than loss) of the mode $\gamma(p) = \mathrm{argmax}_{\omega \in \Omega} p(\{\omega\})$, which is elicitable for finite $\Omega$ (but not otherwise; see Heinrich [19]).

In both cases, one can easily check that the level sets of $\gamma$ are not convex, so $\mathsf{elic}_\mathcal{I}(\gamma) = 2$; alternatively Theorem 2 applies in the first case. As mentioned following Definition 6, the result for finite $\Omega$ differs from the definitions of Lambert et al. [17], where the elicitation complexity of $\gamma$ is $|\Omega| - 1$.

## 4.4 Expected Shortfall and Other Spectral Risk Measures

One important application of our results on the elicitation complexity of the Bayes risk is the elicitability of various financial risk measures. One of the most popular financial risk measures is *expected shortfall* $\mathrm{ES}_\alpha : \mathcal{P} \to \mathbb{R}$, also called *conditional value at risk (CVaR)* or *average value at risk (AVaR)*, which we define as follows (cf. [20, eq.(18)], [21, eq.(3.21)]):

$$\mathrm{ES}_\alpha(p) = \inf_{z \in \mathbb{R}} \left\{ \mathbb{E}_p \left[ \tfrac{1}{\alpha}(z - y)\mathbb{1}_{z \geq y} - z \right] \right\} = \inf_{z \in \mathbb{R}} \left\{ \mathbb{E}_p \left[ \tfrac{1}{\alpha}(z - y)(\mathbb{1}_{z \geq y} - \alpha) - y \right] \right\} . \quad (3)$$

Despite the importance of elicitability to financial regulation [11, 22], $\mathrm{ES}_\alpha$ is not elicitable [7]. It was recently shown by Fissler and Ziegel [15], however, that $\mathrm{elic}_\mathcal{I}(\mathrm{ES}_\alpha) = 2$. They also consider the broader class of *spectral risk measures*, which can be represented as $\rho_\mu(p) = \int_{[0,1]} \mathrm{ES}_\alpha(p)d\mu(\alpha)$, where $\mu$ is a probability measure on $[0,1]$ (cf. [20, eq. (36)]). In the case where $\mu$ has finite support $\mu = \sum_{i=1}^k \beta_i \delta_{\alpha_i}$ for point distributions $\delta$, $\beta_i > 0$, we can rewrite $\rho_\mu$ using the above as:

$$\rho_\mu(p) = \sum_{i=1}^k \beta_i \mathrm{ES}_{\alpha_i}(p) = \inf_{z \in \mathbb{R}^k} \left\{ \mathbb{E}_p \left[ \sum_{i=1}^k \frac{\beta_i}{\alpha_i}(z_i - y)(\mathbb{1}_{z_i \geq y} - \alpha_i) - y \right] \right\} . \quad (4)$$

They conclude $\mathrm{elic}_\mathcal{I}(\rho_\mu) \leq k + 1$ unless $\mu(\{1\}) = 1$ in which case $\mathrm{elic}_\mathcal{I}(\rho_\mu) = 1$. We show how to recover these results together with matching lower bounds. It is well-known that the infimum in eq. (4) is attained by any of the $k$ quantiles in $q_{\alpha_1}(p), \ldots, q_{\alpha_k}(p)$, so we conclude $\mathrm{elic}_\mathcal{I}(\rho_\mu) \leq k + 1$ by Theorem 1, and in particular the property $\{\rho_\mu, q_{\alpha_1}, \ldots, q_{\alpha_k}\}$ is elicitable. The family of losses from Corollary 1 coincide with the characterization of Fissler and Ziegel [15] (see § D.1). For a lower bound, as $\mathrm{elic}_\mathcal{I}(\{q_{\alpha_1}, \ldots, q_{\alpha_k}\}) = k$ whenever the $\alpha_i$ are distinct by Lemma 3, Theorem 2 gives us $\mathrm{elic}_\mathcal{I}(\rho_\mu) = k + 1$ whenever $\mu(\{1\}) < 1$, and of course $\mathrm{elic}_\mathcal{I}(\rho_\mu) = 1$ if $\mu(\{1\}) = 1$.

## 4.5 Variantile

The $\tau$-expectile, a type of generalized quantile introduced by Newey and Powell [23], is defined as the solution $x = \mu_\tau$ to the equation $\mathbb{E}_p \left[ |\mathbb{1}_{x \geq y} - \tau|(x - y) \right] = 0$. (This also shows $\mu_\tau \in \mathcal{I}_1$.) Here we propose the $\tau$-*variantile*, an asymmetric variance-like measure with respect to the $\tau$-expectile: just as the mean is the solution $x = \mu$ to the equation $\mathbb{E}_p[x - y] = 0$, and the variance is $\sigma^2(p) = \mathbb{E}_p[(\mu - y)^2]$, we define the $\tau$-variantile $\sigma_\tau^2$ by $\sigma_\tau^2(p) = \mathbb{E}_p \left[ |\mathbb{1}_{\mu_\tau \geq y} - \tau|(\mu_\tau - y)^2 \right]$.

It is well-known that $\mu_\tau$ can be expressed as the minimizer of a *asymmetric least squares* problem: the loss $L(x, y) = |\mathbb{1}_{x \geq y} - \tau|(x - y)^2$ elicits $\mu_\tau$ [23, 7]. Hence, just as the variance turned out to be a Bayes risk for the mean, so is the $\tau$-variantile for the $\tau$-expectile:

$$\mu_\tau = \operatorname*{argmin}_{x \in \mathbb{R}} \mathbb{E}_p \left[ |\mathbb{1}_{x \geq y} - \tau|(x - y)^2 \right] \implies \sigma_\tau^2 = \min_{x \in \mathbb{R}} \mathbb{E}_p \left[ |\mathbb{1}_{x \geq y} - \tau|(x - y)^2 \right] .$$

We now see the pair $\{\mu_\tau, \sigma_\tau^2\}$ is elicitable by Corollary 1, and by Theorem 2 we have $\mathrm{elic}_\mathcal{I}(\sigma_\tau^2) = 2$.

## 4.6 Deviation and Risk Measures

Rockafellar and Uryasev [21] introduce "risk quadrangles" in which they relate a risk $\mathcal{R}$, deviation $\mathcal{D}$, error $\mathcal{E}$, and a statistic $\mathcal{S}$, all functions from random variables to the reals, as follows:

$$\mathcal{R}(X) = \min_C\{C + \mathcal{E}(X - C)\}, \quad \mathcal{D}(X) = \min_C\{\mathcal{E}(X - C)\}, \quad \mathcal{S}(X) = \operatorname*{argmin}_C\{\mathcal{E}(X - C)\} .$$

Our results provide tight bounds for many of the risk and deviation measures in their paper. The most immediate case is the *expectation quadrangle* case, where $\mathcal{E}(X) = \mathbb{E}[e(X)]$ for some $e : \mathbb{R} \to \mathbb{R}$. In this case, if $\mathcal{S}(X) \in \mathcal{I}_1(\mathcal{P})$ Theorem 2 implies $\mathrm{elic}_\mathcal{I}(\mathcal{R}) = \mathrm{elic}_\mathcal{I}(\mathcal{D}) = 2$ provided $\mathcal{S}$ is non-constant and $e$ non-linear. This includes several of their examples, e.g. truncated mean, log-exp, and rate-based. Beyond the expectation case, the authors show a Mixing Theorem, where they consider

$$\mathcal{D}(X) = \min_C \min_{B_1,..,B_k} \left\{ \sum_{i=1}^k \lambda_i \mathcal{E}_i(X - C - B_i) \,\Big|\, \sum_i \lambda_i B_i = 0 \right\} = \min_{B_1',..,B_k'} \left\{ \sum_{i=1}^k \lambda_i \mathcal{E}_i(X - B_i') \right\} .$$

Once again, if the $\mathcal{E}_i$ are all of expectation type and $\mathcal{S}_i \in \mathcal{I}_1$, Theorem 1 gives $\mathrm{elic}_\mathcal{I}(\mathcal{D}) = \mathrm{elic}_\mathcal{I}(\mathcal{R}) \leq k + 1$, with a matching lower bound from Theorem 2 provided the $\mathcal{S}_i$ are all independent. The Reverting Theorem for a pair $\mathcal{E}_1, \mathcal{E}_2$ can be seen as a special case of the above where

one replaces $\mathcal{E}_2(X)$ by $\mathcal{E}_2(-X)$. Consequently, we have tight bounds for the elicitation complexity of several other examples, including superquantiles (the same as spectral risk measures), the quantile-radius quadrangle, and optimized certainty equivalents of Ben-Tal and Teboulle [24].

Our results offer an explanation for the existence of regression procedures for some of these risk/deviation measures. For example, a proceedure called *superquantile regression* was introduced in Rockafellar et al. [25], which computes spectral risk measures. In light of Theorem 1, one could interpret their procedure as simply performing regression on the $k$ different quantiles as well as the Bayes risk. In fact, our results show that any risk/deviation generated by mixing several expectation quadrangles will have a similar procedure, in which the $B_i'$ variables are simply computed along side the measure of interest. Even more broadly, such regression procedures exist for *any* Bayes risk.

## 5 Discussion

We have outlined a theory of elicitation complexity which we believe is the right notion of complexity for ERM, and provided techniques and results for upper and lower bounds. In particular, we now have tight bounds for the large class of Bayes risks, including several applications of note such as spectral risk measures. Our results also offer an explanation for why procedures like superquantile regression are possible, and extend this logic to all Bayes risks.

There many natural open problems in elicitation complexity. Perhaps the most apparent are the characterizations of the complexity classes $\{\Gamma : \mathsf{elic}(\Gamma) = k\}$, and in particular, determining the elicitation complexity of properties which are known to be non-elicitabile, such as the mode [19] and smallest confidence interval [18].

In this paper we have focused on elicitation complexity with respect to the class of identifiable properties $\mathcal{I}$, which we denoted $\mathsf{elic}_\mathcal{I}$. This choice of notation was deliberate; one may define $\mathsf{elic}_\mathcal{C} := \min\{k : \exists \hat{\Gamma} \in \mathcal{E}_k \cap \mathcal{C}, \exists f, \Gamma = f \circ \hat{\Gamma}\}$ to be the complexity with respect to some arbitrary class of properties $\mathcal{C}$. Some examples of interest might be $\mathsf{elic}_\mathbb{E}$ for expected values, of interest to the prediction market literature [8], and $\mathsf{elic}_{\mathrm{cvx}}$ for properties elicitable by a loss which is convex in $r$, of interest for efficiently performing ERM.

Another interesting line of questioning follows from the notion of *conditional elicitation*, properties which are elicitable as long as the value of some other elicitable property is known. This notion was introduced by Emmer et al. [11], who showed that the variance and expected shortfall are both conditionally elicitable, on $\mathbb{E}_p[y]$ and $q_\alpha(p)$ respectively. Intuitively, knowing that $\Gamma$ is elicitable conditional on an elicitable $\Gamma'$ would suggest that perhaps the pair $\{\Gamma, \Gamma'\}$ is elicitable; Fissler and Ziegel [15] note that it is an open question whether this joint elicitability holds in general. The Bayes risk $\underline{L}$ for $\Gamma$ is elicitable conditioned on $\Gamma$, and as we saw above, the pair $\{\Gamma, \underline{L}\}$ is jointly elicitable as well. We give a counter-example in Figure 1, however, which also illustrates the subtlety of characterizing all elicitable properties.

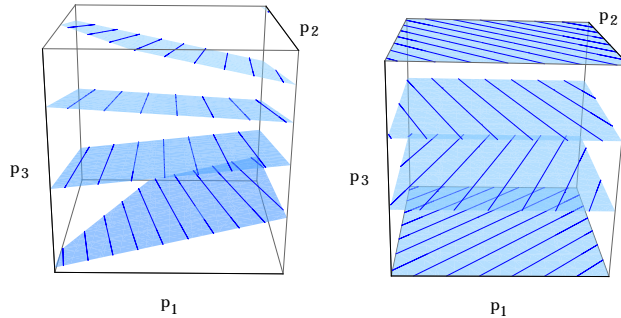

Figure 1: Depictions of the level sets of two properties, one elicitable and the other not. The left is a Bayes risk together with its property, and thus elicitable, while the right is shown in [3] not to be elicitable. Here the planes are shown to illustrate the fact that these are both conditionally elicitable: the height of the plane (the intersept $(p_3, 0, 0)$ for example) is elicitable from the characterizations for scalar properties [9, 1], and conditioned on the plane, the properties are both linear and thus links of expected values, which are also elicitable.

## Footnotes

[1] We will also consider $\Gamma : \mathcal{P} \to \mathbb{R}^{\mathbb{N}}$.

[2]Our main lower bound (Thm 2) merely requires $\Gamma$ to have convex level sets, which is necessary by Prop. 1.

[3]One may take for example $\Gamma(p) = \mathrm{argmax}_i\, p(A_i)$ for a finite measurable partition $A_1, \ldots, A_n$ of $\Omega$.

[4]Note that these restrictions on $\Omega$ may easily be placed on $\mathcal{P}$ instead; e.g. finite $\Omega$ is equivalent to $\mathcal{P}$ having support on a finite subset of $\Omega$, or even being piecewise constant on some disjoint events.

[5]Here and throughout, when $\Omega = \mathbb{R}^k$ we assume the Borel $\sigma$-algebra.

[6]Omitted proofs can be found in the appendix of the full version of this paper.

[7]As we focus on elicitation complexity, we have not tried to characterize all ways to elicit this joint property, or other properties we give explicit losses for. See § 4.1 for an example where additional losses are possible.

[8]Note that one could easily lift the requirement that $\Gamma$ be a function, and allow $\Gamma(p)$ to be the set of minimizers of the loss (cf. [18]). We will use this additional power in Example 4.4.

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
