[Reviews · NeurIPS 2015]

Submitted by Assigned_Reviewer_1

Note: this is a "heavy" review, although I do not have too many comments for the authors in this case. I may have more comments after the discussion with other reviewers.

1) In the introduction it should be made clearer when discussing conditional elicitation if the conditioning property must be known exactly, or if approximately knowing it is enough. This seems to be an important distinction to make, on the face of it.

2) p2: in Definition 1, it's not really clear what is meant by "correct" here -- \Gamma seems to need no notion of "correctness" -- it's just a function from distributions to parameter values. Obviously one could construct some weird examples of non-elicitable functions, but the correctness has to do with whether something is elicitable or not, right?

3) The notion of complexity here is about the number of parameters or dimension of the parameter space, which seems a bit different than e.g. VC dimension or other ideas about complexity. I think this overloading of the word may confuse readers.

4) The mean/variance/second moment example makes a lot of the setup crystal clear and is accessible to most readers. I think putting this earlier in the paper, say even in the introduction, could help a lot.

5) Must \mathcal{A} in Theorem 1 be finite? Are there any restrictions on this set?

6) The discussion of conditional elicitation seems rather rushed and cursory. If the authors intend to produce a journal version of this work then I would, for the NIPS version, remove this part and expand the examples and discussion a bit more. However, that is just a suggestion.

Small things: - In section 3, check the grammar.

AFTER AUTHOR FEEDBACK: I think moving Section 4 to a journal version and putting more examples would be great. The variance example was very helpful and having it earlier would be much better.

I'm not changing my score, but I think trying to position the paper to be more NIPS-friendly would definitely help.
Summary: This is a very nice paper on the problem of elicitation, or how to compute functionals of distributions from samples (using ERM). Some properties are elicitable using ERM. Some are not. This paper gives a more nuanced view by asking about the complexity of elicitation -- that is, how complex a functional (or how many parameters) can one elicit from a distribution using ERM alone?

Submitted by Assigned_Reviewer_2

The paper provides various results based on a new definition of election complexity (definition 6). Overall I am not qualified to judge the importance of this, hence my low confidence score. I can say that the paper is remarkably easy to follow (up to a point) given my unfamiliarity with the area. The writing is good, and the logical thread is fairly clear. I did spend some time thinking about the theorems and couldn't find any obvious problems.

I liked theorem 1. Are there similar results in the literature?

A few minor points:

What is the point of proposition 3? It seems like an obfuscated way of saying something fairly obvious and doesn't seem to connect to the main thread.

Line 279: I assume this should be L((x,v),y) not L(x,v,y)

Line 330: missing d in generalized
Summary: The author proposes a new definition of elicitation complexity based on a link function along with some consequences. There is also a section on conditional elicitation in a necessary condition is given for identifiability to imply joint elicit ability.

Submitted by Assigned_Reviewer_3

This paper introduces an adapted definition of elicitation complexity and shows the upper and lower bound of the complexity. The paper is well-written, well-referenced, and presents a solid technical contribution, though simple proof techniques.

Specific comments: - As for conditional elicitation, the theorem is derived based on a twice differentiable assumption, how about the results on other loss function cases? - It would be better to provide some empirical comparison results.
Summary: A solid and promising contribution, worthy of publication

Submitted by Assigned_Reviewer_4

=======POST REBUTTAL AND DISCUSSION COMMENTS: I seem to be in the minority in thinking the paper is too technical for the NIPS crowd. I am raising my score accordingly. =======

This paper considers the problem of property elicitation. A property of a probability distribution is simply a (scalar) function of this distribution, and a property G is said to be elicitable if there exists a loss function such that the minimizer of the expected loss for a distribution P coincides with G(P). The authors observe that the ability to elicit a function of a distribution is something of a weak notion, since one can always elicit the full distribution and then return the property value. Instead, the important question is to ask *how many* parameters would need to be elicited in order to produce the property value for all distributions. The paper provides some nice results that help to understand the notion of "elicitation complexity", and the paper gives some applications of these results that connect with recent work. The paper also considers a new notion of "conditional elicitation" and provides some limited exploration of this idea.

On the positive side, the paper does seem to provide nice novel results that help to understand elicitation, making nice connections with the estimation of the Bayes risk. The framework the authors put forth is coherent and natural, and builds upon a great deal of previous but recent work on this topic. The paper really does bring out a coherent way to reason about elicitability. There seem to be cool tricks in here too; in particular, I noticed that Theorem 1 characterizes a class of scoring rules that require an increasing function, a tool I've not seen previously.

On the negative side, and why I think this paper is really going to be a stretch for NIPS, is that: (a) the work is highly technical, (b) the paper would be readable by only a small subset of the NIPS community, (c) the technical parts are quite dense with very little in the way of examples or helpful discussion, (d) the 8-page format is too small to lay out the framework appropriately, with motivating examples etc. Overall i think the authors really aren't doing justice to this work if they want to sell this to the broader ML community. This paper, after expanding and improving the technical content, would be much more appropriate for COLT or a statistics journal. I'm probably a 5.5 on this paper, but I could be swayed based on the response and the feelings of the other reviewers.

More detailed comments: -- The authors use terms like "how much effort is required to compute it via ERM" but I think this confuses the notion of effort. The "complexity" discussed in this paper is more about description length and not about computational complexity or sample complexity. Those are also interesting questions but not really addressed. -- The beginning of Section 2 is way too dense. The reader not familiar with this line of work is going to find this impossible to wade through. -- Why does 'elic' have a subscript of \mathcal{I} but 'iden' does not? -- The reason that elic(\Gamma) >= iden(\Gamma) should be done much more explicitly, I still don't understand why it's obviously true. -- The authors appear to combine properties via {}-notation, but {} are usually used for sets and so this is confusing. -- I found Theorem 2 quite hard to parse. -- footnote 7 has Example 3.1.4 mentioned twice. -- On the one hand, section 4 seems very dense and quite rushed. On the other hand, there's space at the bottom of page 8 which suggests there was room to add more explanation.
Summary: There are nice results here, and the paper introduces a very nice framework for reasoning about elicitation. But the paper is quite dense, the notation very heavy, and I expect that only a handful of folks from the NIPS community will find this paper interesting.

Author Feedback
Author rebuttal: We thank the reviewers for their feedback, and want to begin by addressing the issue that several raised about the technical nature of the presentation (particularly in Section 2) and the resulting implications for the accessibility and interest of the work to the broad NIPS audience. First, we do think that our main object of study, the number of parameters needed to learn a particular function using ERM, is something with broad interest, and not restricted to, e.g., the COLT subcommunity. However, in our submitted version we definitely agree that we struggled to make the somewhat technical nature of the precise definitions compatible with an accessible presentation. One of the reviewers has a very helpful suggestion in this regard, that of moving the variance example earlier, and we agree that this would definitely help motivate and clarify the definitions. Additionally, we do plan to produce a journal version of this paper, which does leave us flexibility in terms of which material gets included in a conference version. If the reviewers feel that it would be more valuable to the NIPS audience to save the material in Section 4 for a journal version and use the space for more examples and discussion earlier in the paper, that is an idea we are very open to.

There are a number of suggestions and corrections we agree with and plan to enact, but also others we wanted to comment on and questions we wanted to answer:

Q: "As for conditional elicitation, the theorem is derived based on a twice differentiable assumption, how about the results on other loss function cases?"

A: We believe this is true more generally, but the tools used in our proof (based on prior work) impose this restriction. Also, note that our upper bound results (e.g. Thm 1) apply in this case without restriction.

Q: Why elicitation "complexity"?

A: This term is consistent with prior work (e.g. [17]). We agree it is potentially confusing, but other terms (e.g. "elicitation order" from [15], which has also been used to refer to the dimension of the property/statistic) do not seem like an improvement. We would be open to any suggestions.

Q: "Why does 'elic' have a subscript of \mathcal{I} but 'iden' does not?"

A: The subscript I indicates that the property linked to (\hat\Gamma) must additionally be identifiable, so it would be redundant on iden.

Q: The reason that elic(\Gamma) >= iden(\Gamma)?

A: The statement made is elic_I(\Gamma) >= iden(\Gamma), and so hopefully the answer to the above question now makes this clearer: we have defined elic_I to be those properties which are links of elicitable and identifiable properties, whereas iden only requires links of identifiable properties. More formally, elic_I(\Gamma)=k means \Gamma is a link of \hat\Gamma which is in both I_k and E_k, and since \hat\Gamma is in I_k, we can immediately conclude iden(\Gamma) <= k.

Q: "In the introduction it should be made clearer when discussing conditional elicitation if the conditioning property must be known exactly, or if approximately knowing it is enough."

A: Our results require knowing it exactly, but is an interesting question of when knowing it approximately would be enough in practice. We will certainly clarify.

Q: "in Definition 1, it's not really clear what is meant by "correct" here -- \Gamma seems to need no notion of "correctness" -- it's just a function from distributions to parameter values. Obviously one could construct some weird examples of non-elicitable functions, but the correctness has to do with whether something is elicitable or not, right?"

A: Exactly right; thank you for flagging this term (a holdover from the economics/crowdsourcing literature) as confusing.

Q: "Must \mathcal{A} in Theorem 1 be finite? Are there any restrictions on this set?"

A: No, because we are just using a pointwise supremum, which leads to a convex function for arbitrary A. The theorem does require the existence of a minimizer, for which finiteness is of course sufficient, but strictly speaking this is not a restriction on A.

Q: "I liked theorem 1. Are there similar results in the literature?"

A: The result in [15] is a special case (where an increasing function appears in their characterization as well, as noted by R3). Section 3.1.6 also outlines some similar results in [23] for risk measures.

Q: "What is the point of proposition 3? It seems like an obfuscated way of saying something fairly obvious and doesn't seem to connect to the main thread."

A: The connection is in establishing that there is a general upper bound. We agree it may be obvious for people with some analysis background, but felt it important to note in any case.